# Multi-level Protein Structure Pre-training with Prompt Learning

**Zeyuan Wang**[1,2,7*]   **Qiang Zhang**[1,2*†]   **Haoran Yu**[2,3]   **Shuangwei Hu**[4]   **Xurui Jin**[5]
**Zhichen Gong**[2,6]   **Huajun Chen**[1,2, 7, 8†]
[1]College of Computer Science and Technology, Zhejiang University
[2]ZJU-Hangzhou Global Scientific and Technological Innovation Center
[3]College of Chemical and Biological Engineering, Zhejiang University
[4]Vecx Biomedicines Inc., [5]MindRank AI Ltd., [6]University College London
[7]AZFT Joint Lab for Knowledge Engine, [8]East China Sea Laboratory
{yuanzew,qiang.zhang.cs,yuhaoran,huajunsir}@zju.edu.cn
shuangwei@vecx.bio, xurui@mindrank.ai, ucabzgo@ucl.ac.uk

## Abstract

A protein can focus on different structure levels to implement its functions. Each structure has its own merit and driving forces in describing specific characteristics, and they cannot replace each other. Most existing function prediction methods take either the primary or the tertiary structure as input, unintentionally ignoring the other levels of protein structures. Considering protein sequences can determine multi-level structures, in this paper, we aim to realize the comprehensive potential of protein sequences for function prediction. Specifically, we propose a new prompt-guided multi-task pre-training and fine-tuning framework. Through the prompt-guided multi-task pre-training, we learn multiple prompt signals to steer the model, called PromptProtein, to focus on different levels of structures. We also design a prompt fine-tuning module to provide downstream tasks the on-demand flexibility of utilizing respective levels of structural information. Extensive experiments on function prediction and protein engineering show that PromptProtein outperforms state-of-the-art methods by large margins. To the best of our knowledge, this is the first prompt-based pre-trained protein model.

## 1 Introduction

Pre-trained language models (PTLMs) have prevailed in natural language processing (NLP). Recently, some methods (Alley et al., 2019; Elnaggar et al., 2021; Rives et al., 2021) use PTLMs to encode protein sequences to predict biological functions, which are called pre-trained protein models (PTPMs). In contrast to natural languages, there are four distinct levels of protein structures (Kessel & Ben-Tal, 2018). The primal is the protein sequence consisting of amino acids, the second refers to the local folded structures (e.g., $\alpha$ helix and $\beta$ pleated sheet), the tertiary describes the natural folded three-dimensional structure, and the quaternary is a protein multimer comprising multiple polypeptides. A protein can focus on different structure levels to implement its specific functions, including reserving a piece of the sequence, manifesting the whole 3D structure as conformational elements, or even cooperating with other proteins. Therefore, when predicting protein functions, it is vital to flexibly utilize multi-level structural information.

AlphaFold2 (Jumper et al., 2021) makes great progress in the tertiary structure prediction based on protein sequences. However, directly learning from predicted structures can be unachievable as the prediction of proteins without homologous sequences is inaccurate. More importantly, the quaternary structure of protein multimers which faithfully depicts protein functions is usually different from the tertiary (see Figure 1) and reliable predictive models have not been released. Fortunately, protein sequences are easy to obtain and can determine all the other levels of structures. This paper aims to realize the full potential of protein sequences in function prediction by prompting a

---

*Equal contribution and shared co-first authorship.
†Corresponding author.

PTPM to exploit all levels of protein structures during pre-training. The main challenges are two-fold: 1) **how to design proper pre-training tasks for different protein structures**? and 2) **how to efficiently integrate these tasks in the pre-training phase and transfer the implicit protein structure knowledge for function prediction in fine-tuning phase**.

For the first challenge, we design three complementary pre-training tasks across multiple structure levels, targeting both fine and coarse resolutions. Specifically, we use the *de facto* Mask Language Modeling (MLM) task to exploit the primary structure information, where the model needs to predict randomly masked amino acids in a protein. For the secondary and tertiary structure, we propose the alpha-carbon CooRDinate prediction (CRD) task, where the model should output the relative positions between residues. For the quaternary structure, we propose the Protein-Protein Interaction prediction (PPI) task, where the model is required to estimate the interaction probability. We collect millions of data covering different levels of protein structures from UniRef50 (Consortium, 2021), Protein Data Bank (Berman et al., 2000), and STRING (Szklarczyk et al., 2019).

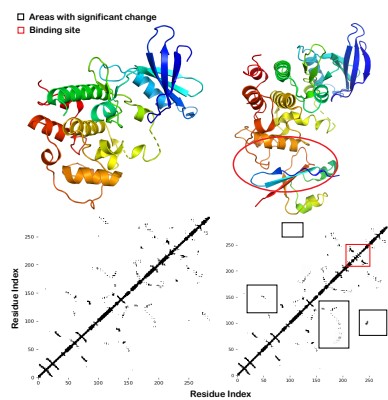

Figure 1: A comparison of protein CDK1 in the tertiary (**left**) and quaternary (**right**) structures.

For the second challenge, a straightforward strategy is to leverage multi-task learning to combine the losses of different pre-training tasks. However, many works (Wu et al., 2019; Yu et al., 2020) find that task interference is common when tasks are diverse. This problem can be more severe in multi-task pre-training due to the gap between pre-training and downstream tasks, causing negative knowledge transfer. For example, BERT (Kenton & Toutanova, 2019) leverages MLM and Next Sentence Prediction (NSP) to learn the sequential dependency and sentence relationship simultaneously, while RoBERTa (Liu et al., 2019) finds the performance will be slightly improved when removing the NSP loss. We postulate this problem also exists in multi-level protein structures, as different structures can be inconsonant. The MLM task emphasizes the neighboring relations along the sequence, while the CRD task shall focus more on long-range amino acid pairs which can be spatially close in the tertiary structure.

To address this challenge, inspired by recent prompt learning, we propose a prompt-guided multi-task pre-training and fine-tuning framework, and the resulting protein model is called PromptProtein. The prompt-guided multi-task pre-training associates multiple pre-training tasks with dedicated sentinel tokens, called prompts. To utilize the prompt tokens, we introduce a prompt-aware attention module, which modifies two components of the Transformer architecture: 1) Attention mask, which is designed to block attention calculation from input data to a prompt as a prompt should be task-dependent instead of sample-dependent. 2) For skip connection, a prompt is used to calculate a skip weight, which can filter out task-irrelevant information. At the fine-tuning phase, we propose a prompt fine-tuning module to coordinate all prompt tokens, such that the model is capable of leveraging multi-level protein structure information flexibly, enabling the positive transfer of learned structural knowledge to downstream tasks.

We conduct experiments on function prediction and protein engineering as downstream tasks, where PromptProtein significantly outperforms state-of-the-art on all datasets, especially on low-resource protein engineering tasks where PromptProtein achieves an average improvement of 17.0%.

## 2 RELATED WORKS

**Protein Representation Models.** Proteins have complex structures that determine their biological functions (Epstein et al., 1963). A growing body of work focuses on how to leverage structural information. Since evolution through natural selection has spoken protein sequences as their "natural language", various natural language processing methods have been extended to proteins. Asgari & Mofrad (2015); Yang et al. (2018) apply word embedding algorithms (Mikolov et al., 2013) to obtain protein representations. Dalkiran et al. (2018); Öztürk et al. (2018) use one-dimensional con-

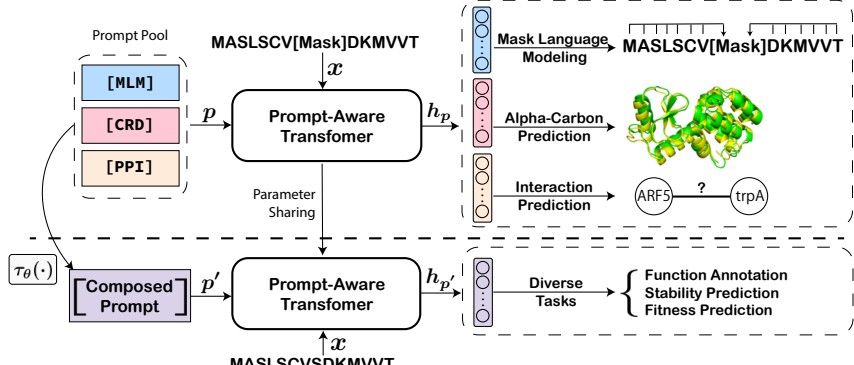

Figure 2: The architecture overview of PromptProtein. In the pre-training stage, we pre-train our model with three structure-related tasks, including mask language modeling, alpha-carbon prediction, and protein-protein interaction prediction. For each task, the model takes the protein sequence and the task-specific token as input and learns to produce a representation encoding the corresponding structure information. In the fine-tuning stage, a prompt-tuning module $\tau_\theta(\cdot)$ can flexibly combine structure information via the learned prompt tokens for diverse downstream tasks.

volutional neural networks to predict the functions. Furthermore, Alley et al. (2019); Elnaggar et al. (2021); Rives et al. (2021) explore whether the pre-training and fine-tuning paradigm, the transformer architectures, and the objective functions can effectively transfer from natural languages to proteins. Zhang et al. (2021a) align the amino acid sequence and the text sequence to obtain informative protein representation. To utilize the tertiary structure, Hermosilla et al. (2020); Somnath et al. (2021); Ganea et al. (2021); Zhang et al. (2022) build protein graphs and employ message-passing neural networks to produce structure-aware representations. Bepler & Berger (2021) employ contact map prediction and structural similarity prediction to pre-train the protein model. Although primary and tertiary structures have been studied, few works try to enrich protein representation with the quaternary structure which faithfully depicts protein functions. In this paper, we show that systematic modeling and flexible utilization of multi-level structures are the keys to improving the performance of function prediction and protein engineering.

**Multi-task Learning.** The goal of multi-task learning is to take advantage of inductive transfer across tasks and achieve better generalization performance. When tasks are diverse, using a naive shared MTL model can suffer from task interference. Prior methods have been proposed to de-conflict gradients from different tasks. Chen et al. (2018) dynamically adjust gradient magnitudes so different tasks can be trained at similar scales. Yu et al. (2020) take the gradient direction into account and drop the projection of one task gradient direction onto another if they are conflicting. Rather than clipping the conflict gradient direction, Javaloy & Valera (2021) learn a rotation matrix for each task to bring different optima closer to each other. However, these methods are not designed for multi-task pre-training and cannot properly deal with the knowledge transferability to downstream tasks. We provide a schematic comparison of these methods in Appendix A.1.

**Prompts for Pre-trained Models.** In-context learning (Brown et al., 2020) is introduced to steer the pre-trained model to produce task-desired representations. In the NLP area, the prevailing approaches to designing prompts can be divided into two categories: discrete prompt designing and continuous prompt tuning. The discrete prompt technique (Schick & Schütze, 2021) adds task description tokens from a vocabulary to the context to obtain enriched sentence embeddings. However, the hand-crafted prompts may provide disturbance of human bias and are limited to discrete vocabulary spaces. In contrast, Li & Liang (2021); Zhang et al. (2021b) generate optimal prompt vectors in continuous spaces. Inspired by these works, we extend the concept of prompt tuning to the pre-training stage, associate multi-level protein structural information with dedicated prompt tokens during pre-training, and adaptively combine these learned prompts for downstream tasks.

## 3 METHODOLOGY

To acquire multiple information from the input data $\boldsymbol{x}$, conventional multi-task learning usually produces a universal representation $\boldsymbol{h}$. The whole objective can be formulated as a weighted sum of individual task objectives: $\mathcal{L} = \sum_i \alpha_i \mathcal{L}_i(\boldsymbol{h})$, where $\{\alpha_i\}$ are the hyper-parameters to balance these losses. However, multi-level protein structures can be inconsonant: the primary structure focuses

more on the dependency along the sequence, whereas the tertiary and quaternary structure weights more on the spatial organization, which can cause the problem of task interference. This problem can lead to more severe negative transfer in multi-task pre-training due to the gap between pre-training and downstream tasks. To solve this problem, we propose a prompt-guided multi-task pre-training and fine-tuning framework that utilizes a prompt token $\boldsymbol{p}$ to produce a task-specific representation $\boldsymbol{h_p}$. Multiple learned tokens can be flexibly combined to steer the pre-trained model for various downstream tasks, bridging the gap between pre-training and downstream tasks.

This section first describes how to use prompts to modify the Transformer architecture, such that different tasks can be processed by different neural layers and reduce task interference. Then we present the three pre-training tasks to acquire multi-level protein structural information: (1) masked language modeling, (2) alpha-carbon coordinate prediction, and (3) protein-protein interaction prediction. Finally, we introduce the prompt-guided pre-training and fine-tuning framework where multiple information can be acquired in the pre-training stage and combined on-demand for downstream tasks. The resulting PromptProtein model is illustrated in Figure 2.

### 3.1 PROMPT-AWARE ATTENTION MODULE

To reduce interference between pre-training tasks, we use the prompt token to modify the Transformer architecture so that multiple information can be effectively acquired by the pre-trained model. Specifically, we modify two parts of the Transformer: attention mask and skip connection, and the resulting architecture is called Prompt-aware Transformer. Given an input protein sequence $\boldsymbol{x}$ and a prompt token $\boldsymbol{p}$, we define the whole input $\boldsymbol{x_p}$ denote $\boldsymbol{x_p} = \boldsymbol{x}||\boldsymbol{p}$, where $||$ is concatenation. Let $\boldsymbol{x_p}^i$ be the $i$-th token of the whole input and $\boldsymbol{h_p}^{(l)}$ be the representation of $\boldsymbol{x_p}$ at the $l$-th layer.

**Attention mask.** The conventional self-attention is formulated as: $\text{Attn}(\boldsymbol{h_p}^{(l)}) = \text{Softmax}((QK^T)/\sqrt{d})V$, where $Q$, $K$, and $V$ are the linear projection of $\boldsymbol{h_p}^{(l)}$. Each token in the whole sequence can attend to others at any position which means the condition prompt will be affected by the input sequence. A more reasonable way is to keep only the effect of the prompt on the input sequence and eliminate the reverse effect, as a prompt should be task-dependent instead of sample-dependent. As illustrated in Figure 3, we design an attention mask matrix $\boldsymbol{M}$ to fulfill this requirement. Let $\boldsymbol{M}_{ij}$ denote the $(i,j)$-element of the mask matrix, and we define:

$$\boldsymbol{M}_{ij} = \begin{cases} 0, \boldsymbol{x_p}^i \in \boldsymbol{p} \text{ and } \boldsymbol{x_p}^j \in \boldsymbol{x} \\ 1, \text{others.} \end{cases} \quad (1)$$

**Skip connection.** Skip connection enables deep neural networks easier to train (He et al., 2016). To encourage different tasks to be processed by different layers and reduce task interference, we design a weighted skip connection. That is, the prompt token is used to calculate a weight for the output of the attention module. The whole process can be:

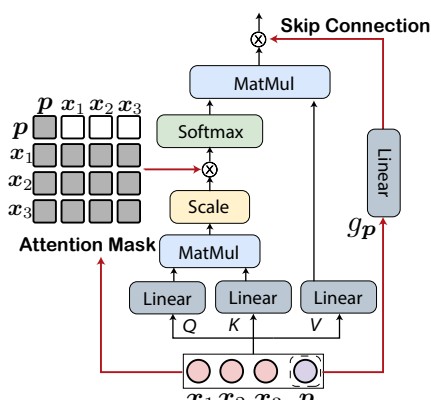

Figure 3: Prompt-aware Attention Module. A pink circle represents an amino acid token and a purple circle represents a prompt token. We decouple prompt tokens from amino acid tokens by the attention mask. The embedding of decoupled prompt token determines the weight of the residual connection. In the fine-tuning stage, we use a prompt-tuning module $\tau_\theta(\cdot)$ to learn the downstream task-desired composed prompt.

$$\boldsymbol{h_p}^{(l+1)} = \boldsymbol{h_p}^{(l)} + (1 - g_{\boldsymbol{p}}^{(l)})\text{Attn}(\boldsymbol{h_p}^{(l)}), \quad (2)$$

where $g_{\boldsymbol{p}}^{(l)}$, a scalar, is linear projection of $l$-th layer embedding of prompt $\boldsymbol{p}$. After $L$ layers of the prompt-aware attention module, we have the task-specific representation $\boldsymbol{h_p} = \boldsymbol{h_p}^{(L)}$.

### 3.2 PROTEIN MULTI-LEVEL STRUCTURES LEARNING

To acquire multi-level protein structure information, we consider three complementary pre-training tasks: (1) masked language modeling, which has been commonly used by existing PTPMs and can

capture the primary structure information; (2) coordinate prediction, which acquires the secondary and tertiary structure; and (3) interaction prediction, which acquires the quaternary structure.

**Masked language modeling.** This task uses all available amino acid tokens to recover the masked ones. Let $Y$ be the set of masked out tokens, and $\mathcal{V}$ be the vocabulary of amino acid tokens. The MLM loss is formulated:

$$q(y|\boldsymbol{h_p}) = \frac{\exp(p(y|\boldsymbol{h_p}))}{\sum_{v\in\mathcal{V}}\exp(p(v|\boldsymbol{h_p}))}, \quad \mathcal{L}_{\text{MLM}}(\boldsymbol{h_p}) = \sum_{y\in Y} -\log q(y|\boldsymbol{h_p}). \quad (3)$$

**Alpha-Carbon Coordinate Prediction.** Since a secondary structure can be inferred from the protein 3D coordinates (Kabsch & Sander, 1983), we use an $\alpha$-C coordinate prediction task to learn both secondary and tertiary structures. Given the sequence length $|\boldsymbol{x}|$, we denote the ground-truth naturally folded 3D structure of protein as $Z \in \mathbb{R}^{|\boldsymbol{x}|\times 3}$ and the structure predictor, a 2-layer MLP network, as $\kappa$, then the predicted structure is $\kappa(\boldsymbol{h_p}) \in \mathbb{R}^{|\boldsymbol{x}|\times 3}$. By translating and rotating (Kabsch, 1976) the predicted structure, we can get the minimal root mean square deviation between ground-truth and predicted structure, and the loss is calculated based on this deviation. In this way, there is no need to consider spatial invariance or equivariance, but only need to focus on the relative positions between residues. The CRD loss can be calculated as the mean square error (MSE):

$$\mathcal{L}_{\text{CRD}}(\boldsymbol{h_p}) = \text{MSE}(Z, \text{Kabsh}(\kappa(\boldsymbol{h_p}))). \quad (4)$$

**Protein-Protein Interaction prediction.** To acquire the quaternary structure information, we conduct the third pre-training task: predicting whether the $m$-th and $n$-th proteins can interact with each other within batched data. Let $\boldsymbol{h_p}^m$ be the $m$-th protein in a mini-batch and $y_{m,n}$ is the ground-truth. We first calculate pair-aware protein representation $\boldsymbol{h_p}^{m,n}$, then formulate the PPI loss:

$$\text{Attn}_{m,n} = \text{Sigmoid}(\frac{(\boldsymbol{h_p}^m W)(\boldsymbol{h_p}^n W)^T}{\sqrt{d}}),$$
$$\boldsymbol{h_p}^{m,n} = \text{mean}(\text{Attn}_{m,n}^T \boldsymbol{h_p}^m)||\text{mean}(\text{Attn}_{m,n}\boldsymbol{h_p}^n)),$$
$$\mathcal{L}_{\text{PPI}}(\boldsymbol{h_p}) = \sum_{m,n\in N} \text{BCE}(y_{m,n}, p(y_{m,n})|\boldsymbol{h_p}^{m,n}), \quad (5)$$

where $W \in \mathbb{R}^{d_W \times d_W}$ is a projection matrix, BCE is the binary cross-entropy loss function, $N$ is the batch size. More details of the pre-training tasks are provided in Appendix A.2.

## 3.3 PROMPT-GUIDED MULTI-TASK PRE-TRAINING AND FINE-TUNING

Corresponding to the three pre-training tasks, the prompt can be instantiated as one of the three tokens, i.e., $\boldsymbol{p} \in P = \{[\text{MLM}], [\text{CRD}], [\text{PPI}]\}$. The task-specific representation is thus denoted as $\boldsymbol{h}_{[\text{MLM}]}, \boldsymbol{h}_{[\text{CRD}]}, \boldsymbol{h}_{[\text{PPI}]}$. The objective function of the prompt-guided multi-task pre-training can be formulated as:

$$\mathcal{L} = \alpha_1 \mathcal{L}_{\text{MLM}}(\boldsymbol{h}_{[\text{MLM}]}) + \alpha_2 \mathcal{L}_{\text{CRD}}(\boldsymbol{h}_{[\text{CRD}]}) + \alpha_3 \mathcal{L}_{\text{PPI}}(\boldsymbol{h}_{[\text{PPI}]}). \quad (6)$$

When we pre-train a model with multiple tasks as Equation 6, both model parameters $\psi$ and prompts $\boldsymbol{p}$ are optimized. In this way, the model does not necessarily need to learn the optimal representation for all tasks, but only needs to learn the respective optimal representation for each task. Hence, the problem of task interference can be alleviated.

Furthermore, to bridge the gap between pre-training and downstream tasks, since the model can acquire each type of information conditioned on the learned prompt tokens, we can combine these tokens with prompt-tuning to flexibly mix the acquired information on-demand. We denote a prompt-tuning module as $\tau_\theta(\cdot)$, and the downstream task-desired protein representation $\boldsymbol{h}_{\boldsymbol{p}'}$ can be obtained by feeding the tuned prompt $\boldsymbol{p}'$

$$\boldsymbol{p}' = \tau_\theta(\boldsymbol{p}_{[\text{MLM}]}, \boldsymbol{p}_{[\text{CRD}]}, \boldsymbol{p}_{[\text{PPI}]}). \quad (7)$$

Then the pre-trained model can produce $\boldsymbol{h}_{\boldsymbol{p}'}$ and conduct predictions for the downstream task of interest. Equation 7 shows how to flexibly utilize the pre-training task information at the fine-tuning stage. Note that, in the pre-training stage, we only append one prompt to acquire one type of task-specific information, while in the fine-tuning stage, we feed all the learned prompt tokens to $\tau_\theta(\cdot)$ and flexibly combine the acquired information. Here, we leverage a linear layer as our prompt-tuning module to combine three learned prompts. For sake of understanding, we provide the pseudo-code of the prompt-guided multi-task pre-training and fine-tuning framework in Appendix A.3.

Table 1: Model performance on EC numbers and GO terms prediction tasks. †: the results taken from Wang et al. (2022), ‡: the results taken from Zhang et al. (2022).

| DATASET | EC | | GO-BP | | GO-MF | | GO-CC | |
|---|---|---|---|---|---|---|---|---|
| | AUPR$_{pair}$ | F$_{max}$ | AUPR$_{pair}$ | F$_{max}$ | AUPR$_{pair}$ | F$_{max}$ | AUPR$_{pair}$ | F$_{max}$ |
| CNN | 0.540 | 0.545 | 0.165 | 0.244 | 0.380 | 0.354 | 0.261 | 0.387 |
| RESNET | 0.137 | 0.187 | 0.166 | 0.280 | 0.281 | 0.267 | 0.266 | 0.403 |
| LSTM | 0.032 | 0.082 | 0.130 | 0.248 | 0.100 | 0.166 | 0.150 | 0.320 |
| TRANSFORMER | 0.187 | 0.219 | 0.135 | 0.257 | 0.172 | 0.240 | 0.170 | 0.380 |
| GAT† | 0.320 | 0.368 | 0.171 | 0.284 | 0.329 | 0.317 | 0.249 | 0.385 |
| GVP† | 0.482 | 0.489 | 0.224 | 0.326 | 0.458 | 0.426 | 0.278 | 0.420 |
| DEEPFRI | 0.547 | 0.631 | 0.282 | 0.399 | 0.462 | 0.465 | 0.363 | 0.460 |
| GearNet − Edge‡ | 0.892 | 0.874 | 0.292 | 0.490 | 0.596 | 0.650 | 0.336 | 0.486 |
| ESM − 1b‡ | 0.889 | 0.864 | 0.343 | 0.470 | 0.639 | 0.657 | 0.384 | 0.488 |
| ProtBERT − BFD† | 0.859 | 0.838 | 0.188 | 0.279 | 0.464 | 0.456 | 0.234 | 0.408 |
| LM − GVP† | 0.710 | 0.664 | 0.302 | 0.417 | 0.580 | 0.545 | 0.423 | 0.527 |
| MT-LSTM | 0.851 | 0.817 | 0.324 | 0.442 | 0.608 | 0.591 | 0.381 | 0.492 |
| MTL | 0.892 | 0.869 | 0.325 | 0.445 | 0.651 | 0.640 | 0.415 | 0.503 |
| GRADNORM | 0.893 | 0.874 | 0.331 | 0.466 | 0.647 | 0.643 | 0.415 | 0.504 |
| ROTOGRAD | 0.895 | 0.876 | 0.334 | 0.470 | 0.648 | 0.638 | 0.416 | 0.509 |
| **PROMPTPROTEIN (OURS)** | **0.915** | **0.888** | **0.363** | **0.495** | **0.665** | **0.677** | **0.457** | **0.551** |

## 4 EXPERIMENTS

### 4.1 PRE-TRAINING SETUP

For the primary structural information, we use UniRef50 (Suzek et al., 2015) which is a clustering of UniRef90 seed sequences at 50% sequence identity. For the secondary and tertiary structural information, we use Protein Data Bank (PDB) (Berman et al., 2000), which includes 200,000 protein 3D structures obtained by experimental methods. For the quaternary structure information, we use the STRING dataset (Szklarczyk et al., 2019) that contains amino acid sequences and protein-protein interaction pairs. In the STRING dataset, protein interactions are divided into 7 categories. We selected the physical-only interaction subset from STRING which contains 65 million protein sequences from 14,095 species and 2.7 billion protein-protein interaction pairs.

We implement PromptProtein using Pytorch (Paszke et al., 2019) and Fairseq (Ott et al., 2019). PromptProtein has 650M parameters with 33 layers and 20 attention heads. The embedding size is 1280. The learning rate is $1 \times 10^{-4}$ with no weight decay. We use an inverse square root learning rate schedule. All models are trained on 2×A100 40G GPUs for 270k steps of updates. After pre-training, the average error of the coordinate prediction task on a single residue is 5 Å, and the accuracy of physical binding prediction is greater than 90.0%. Unless otherwise specified, we use this model in all downstream experiments. The source code will be available online. Please refer to Appendix B for the details of all the pre-training and downstream task dataset statistics.

### 4.2 DOWNSTREAM TASKS: FUNCTION ANNOTATION

**Datasets and Metrics.** Gene ontology (GO) terms and enzyme commission (EC) numbers are two standard classification schemes that organize myriad protein functions. These function prediction tasks can be regarded as multiple binary classification tasks. We follow the dataset split method in (Gligorijević et al., 2021). The evaluation metrics are protein-centric maximum F-score ($F_{max}$) and term-centric area under precision-recall (AUPR) curve, which are used in the CAFA challenges (Radivojac et al., 2013).

**Baselines.** There are four categories of baselines. (1) Sequence-based encoders. CNN (Shanehsazzadeh et al., 2020), ResNet, LSTM, and Transformer (Rao et al., 2019) only take amino acid sequence as input; (2) Geometric learning method. GAT (Veličković et al., 2018), GVP (Jing et al., 2020), DeepFRI (Gligorijević et al., 2021), and GearNet-Edge (pre-trained by Multiview Contrast) (Zhang et al., 2022) take protein 3D coordinates as additional input to obtain informative representation; (3) Pre-trained protein models. ESM-1b (Rives et al., 2021), ProtBERT-BFD (Elnaggar et al., 2021), and LM-GVP (Wang et al., 2022) learn the pattern from large protein corpus. MT-LSTM (Bepler & Berger, 2021) uses contact map and structure similarity to enrich the embed-

Table 2: Model performance on protein engineering tasks. Results with two decimal places are token from Dallago et al. (2021).

| DATASET | STABILITY | FLUORE. | THERMO MIXED | AAV 1-VS-R | 1-VS-R | GB1 2-VS-R | 3-VS-R |
|---|---|---|---|---|---|---|---|
| CNN | 0.51 | 0.67 | 0.34 | 0.48 | 0.17 | 0.32 | **0.83** |
| RESNET | 0.73 | 0.21 | 0.353 | 0.173 | 0.117 | 0.210 | 0.291 |
| LSTM | 0.69 | 0.67 | 0.317 | 0.215 | 0.124 | 0.349 | 0.491 |
| ESM-UNTRAINED | 0.452 | 0.337 | 0.36 | 0.01 | 0.05 | 0.05 | 0.46 |
| ESM-1B | 0.71 | 0.68 | 0.68 | 0.04 | 0.32 | 0.36 | 0.54 |
| ESM-1V | 0.726 | 0.507 | 0.67 | 0.18 | 0.32 | 0.32 | 0.77 |
| PROTBERT-BFD | 0.732 | 0.675 | 0.651 | 0.234 | 0.303 | 0.387 | 0.654 |
| LSTM-MT | 0.741 | 0.648 | 0.665 | 0.258 | 0.335 | 0.402 | 0.741 |
| **PROMPTPROTEIN (OURS)** | **0.767** | **0.683** | **0.694** | **0.551** | **0.403** | **0.550** | 0.783 |

dings. (4) Multi-task learning framework. We employ naive multi-task learning (MTL) and two optimization methods (GradNorm (Chen et al., 2018), RotoGram (Javaloy & Valera, 2021)).

**Results.** We present the evaluation results of proposed PromptProtein and state-of-the-art baselines in Table 1. Compared with all baselines, PromptProtein achieves new state-of-the-art performance on all tasks, which indicates that systematic modeling of multi-level structure information is beneficial. Although the multi-task learning baselines integrate the same information as PromptProtein, they cannot learn multiple information well and transfer properly to downstream tasks. Their inferior performance in GO-BP and GO-CC suggests that there is a gap between downstream task-desired representations and universal pre-trained representations. Flexible composing of structural information significantly improves the performance of the model for downstream tasks.

### 4.3 DOWNSTREAM TASKS: PROTEIN ENGINEERING TASKS

**Datasets and Metrics.** Protein engineering is regarded as a sequence regression task that, given a protein, models are required to identify the functional strength, often termed the fitness landscape. Here, we employ five datasets (stability, fluorescence, thermostability, AAV, and GB1) coming from TAPE (Rao et al., 2019) and FLIP (Dallago et al., 2021) to evaluate whether the model can produce accurate quantitative predictions of these functions. We report the commonly-used Spearman's $\rho$ (rank correlation coefficient) to measure the degree to which the landscape was learned. Results of other tasks on FLIP can be found in Appendix 5.

**Baselines.** For proteins without 3D structures, geometric methods cannot directly apply to these tasks. We choose sequence-based methods (CNN, LSTM, Transformer) and pre-trained protein methods (ESM-1b, ESM-1v (Meier et al., 2021), ProteinBert-BFD, LSTM-MT) as baselines for protein engineering tasks. As Dallago et al. (2021) purport that the various pooling choices perform inconsistently across datasets and splits, for a fair comparison, we utilize the mean pooling method to obtain protein representation.

**Results.** From Table 2, we observe that PromptProtein obtains better performance than all baselines. It confirms that pre-training on structural objectives contributes to protein engineering tasks and systematic modeling of protein multi-level structure leads to further improvements. Note that LSTM-MT, which leverages the tertiary structural information to enhance protein representations, cannot surpass ESM-1b

Table 3: Ablation of PromptProtein with different components.

| METHOD | GB1 | AAV | THERMO |
|---|---|---|---|
| CONVENTIONAL MTL. | 0.238 | 0.525 | 0.651 |
| PROMPTPROTEIN | **0.279** | 0.544 | 0.672 |
| - ATTENTION MASK | 0.264 | 0.531 | 0.663 |
| - LAYER SKIP | 0.270 | 0.520 | 0.659 |
| - MLM OBJECTIVE | 0.240 | 0.493 | 0.629 |
| - CRD OBJECTIVE | 0.262 | 0.535 | 0.647 |
| - PPI OBJECTIVE | 0.253 | 0.532 | 0.654 |

on all datasets, while our proposed approach obtains superior performances. This observation demonstrates that not all structural information leads to positive transfer and flexible utilization of structural information is the key to improved performance. Moreover, PromptProtein can obtain 17.0% improvement on average in low-resource settings of the AAV and GB1 datasets, compared

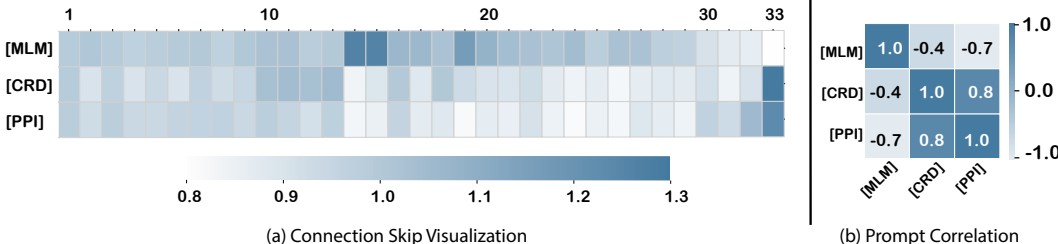

Figure 4: Skip connection visualization and prompt correlation. (a) We visualize the learned skip weight at all neural layers. The darkness of a block represents the weight of that block utilized for the given prompt. (b) We provide the Pearson's correlation between skip weights. The skip patterns between the [MLM] prompt and the other two prompts are negatively correlated, whereas the pattern between the tertiary and quaternary structures is positively correlated.

to the well-performed PTPM baselines. These results indicate that the prompt-guiding PTPM is a better few-shot learner.

## 4.4 ABLATION STUDY

The ablation study is conducted to validate the effectiveness of designed modules in PromptProtein, i.e., prompts, attention mask, or skip connection. As illustrated in Table 3, the performance will decay if any one of the modules is absent, demonstrating that all the modules are advantageous. Furthermore, we notice that skip connection contributes most to the performance, confirming the necessity of reducing task interference.

## 4.5 ANALYSIS AND DISCUSSION

**How do prompts determine the processing pathways of structural information?**

In Figure 4(a), we visualize the skip weights of three pre-trained prompts at different neural layers, and compute the Pearson's correlation (Benesty et al., 2009) of these skip weights to measure the mutual correlations between the pre-training tasks (Figure 4(b)). We have the following key observations. (a) The skip weights are similar in the bottom layers (1-13) across all prompts, indicating all three tasks are processed by these layers. The MLM task information is mainly acquired by the middle layers (14-29), whereas the CRD and PPI information is more acquired by the top layers (30-33). (b) We clearly observe that the [CRD] and [PPI] prompts are more correlated. This is consistent with the intuition that the tertiary and quaternary levels are 3D structures whose amino acids attend to spatially adjacent neighbors, resulting in similar skip weight patterns. Further analysis of the model layer can be found in Appendix B.3.

**Can PromptProtein learn multi-level structures?**

To examine whether prompt-guided pre-training can learn multiple structure information, we conduct experiments to visualize the protein representations conditioned on different pre-trained prompt tokens. We use t-SNE (van der Maaten & Hinton, 2008) to reduce the dimension of embeddings. Figure 5(a) illustrates amino acid embeddings conditioned on [MLM]. We observe that amino acid embeddings in a protein are grouped according to their type. Figure 5(b) illustrates amino acid embeddings conditioned on [CRD]. We find that amino acids are linearly arranged in 2D space along their sequence in the protein. To obtain a more accurate relationship between representations and structures, we compare the protein contact map and the coordinate of embedding. The strong correlation between them demonstrates the CRD objective can effectively learn information about protein 3D structures. In Figure 5(c), we visualize the amino acid embeddings with traditional multi-task pre-training and highlight serine (a class of amino acids). The embeddings attempt to merge multiple structural features at the same time, which leads to an unclear pattern. These results show that prompt-guided pre-training mitigates task interference and allows the multiple structure information to be learned well, resulting in promising performance.

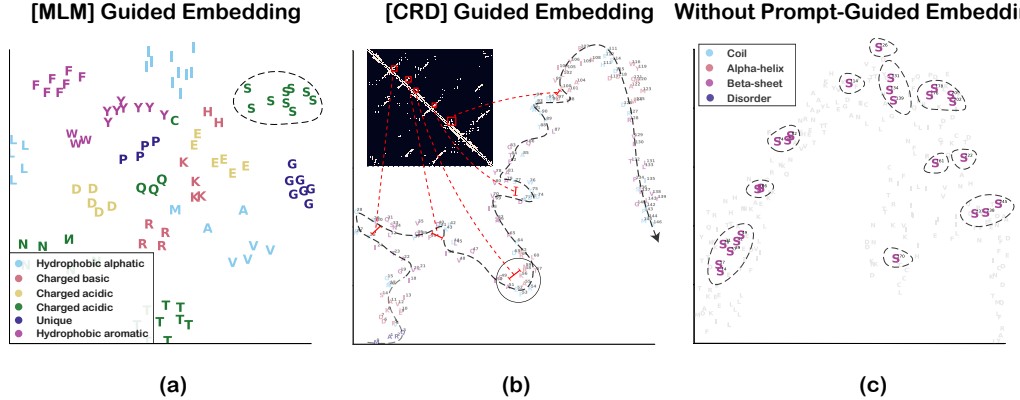

**(a)**      **(b)**      **(c)**

Figure 5: The comparison of amino acid embeddings with different learning methods. We visualize protein representations from prompt-guided multi-task pre-training in (a) and (b), and naive multi-task learning in (c). Each letter represents an amino acid and is colored according to the physicochemical properties in (a), and the secondary structure types in (b) and (c). The superscripts of letters represent the sequential number of amino acids from the C-terminal to the N-terminal.

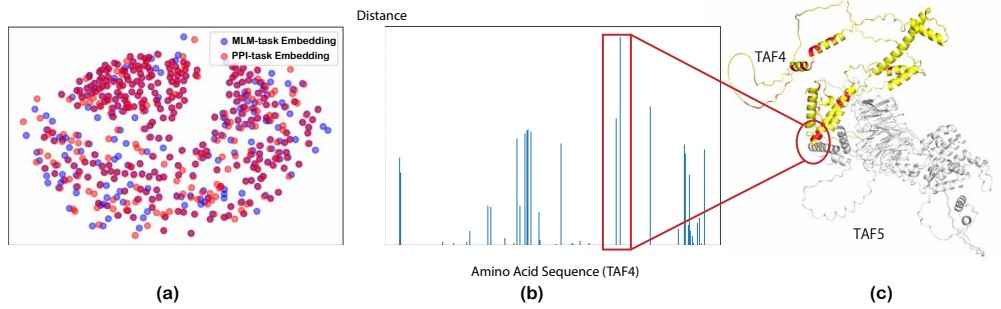

**(a)**      **(b)**      **(c)**

Figure 6: Visualization of the difference of [MLM] and [PPI] prompts. The two proteins are Transcription initiation factor TFIID subunit4 (TAF4) and Transcription initiation factor TFIID subunit 5 (TAF5). **Left**: Visualize the embedding of amino acids conditioned on [MLM] and [PPI] prompts (TAF4) by MDS. **Middle**: Visualize distances of corresponding amino acids in (a). **Right**: Visualize the amino acids with the most variation embeddings (red).

Furthermore, since the [PPI] prompt is trained to provide quaternary structure information, we analyze what exactly the amino acid representations have changed. As shown in Figure 6(a), we firstly visualize the embeddings of amino acids of the TAF4 protein conditioned on [MLM] or [PPI] based on MDS (Kruskal, 1964). Then we calculate the distances between two embeddings of the same amino acid and plot them in Figure 6(b). We mark 30 amino acids with the most variation embeddings in red (Figure 6(c)). The observation that marked amino acids are all on the protein surface is consistent with the fact that modeling the quaternary structure cares about the surface conformation, not the core (Yan et al., 2008).

## 5    CONCLUSION AND FUTURE WORK

In this paper, we extend the concept of prompts from NLP to protein representations. We propose the prompt-guided multi-task pre-training and fine-tuning framework. With this framework, we propose three complementary pre-training structures to acquire multi-level structure information, and flexibly combine them for various downstream tasks. Experimental results on function prediction and protein engineering show that the proposed approach can produce satisfactory improvements when compared to the conventional PTPMs. The improvement is especially significant in low-resource settings. In the future, we are interested in examining the effectiveness of the proposed prompt-guided multi-task pre-training and fine-tuning framework in domains where hierarchical task information is required in the pre-training stage.

ACKNOWLEDGMENTS

This work is funded by NSFC91846204/U19B2027 and sponsored by CAAI-Huawei MindSpore Open Fund. We want to express gratitude to the anonymous reviewers for their hard work and kind comments and Hangzhou AI Computing Center for their technical support. Xurui Jin is the employee of the MindRank AI Ltd.

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

# A    MORE DETAILS OF PROMPTPROTEIN

## A.1    PROMPT-GUIDED MULTI-TASK PRE-TRAINING

One of the key problem to multi-task learning is what to share. Naive and gradient-based methods try to learn a shared MTL model. To overcome between task interference between tasks, they adjust magnitude and direction of gradients . However, negative transfer between pre-training and downstream tasks cannot be mitigated. To realize the potential of positive transfer, multi-task pre-training requires to learn and use task differences on-demand. Both adapter-based approaches and our proposed prompt-based approaches can learn task differences, whereas, for the flexibility of input, only prompt-based approach can use them on-demand. Figure 7 compares the mentioned multi-task methods.

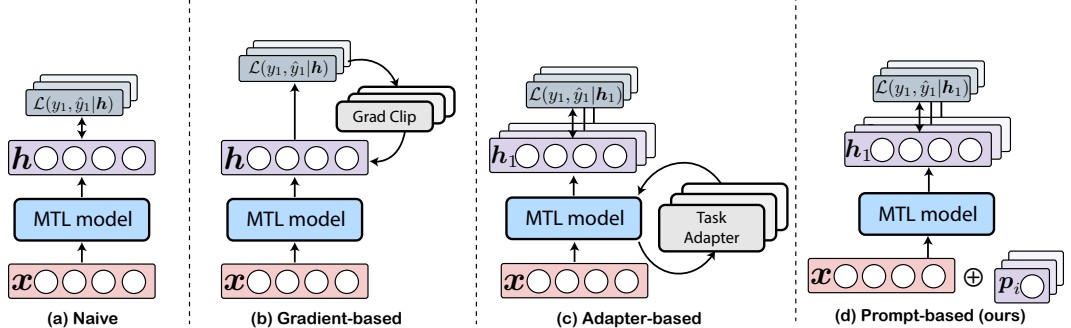

Figure 7: Comparison of multi-task pre-Training.

## A.2    PRE-TRAINING TASKS

In Figure 8, we illustrate our proposed two additional pre-training tasks.

**Alpha-Carbon Coordinate Prediction.** We use a MLP network to project protein embeddings into 3D space. To equip the model with 3D invariance, after predicting the protein coordinates, we first recenter the ground-truth coordinate $Z$ and predicted coordinate $\hat{Z}$ and then employ Kabsch algorithm (Kabsch, 1976) to calculate the optimal rotation matrix that minimizes the root mean squared deviation. We first calculate cross-covariance matrix between two sets of coordinates:$H = Z^T\hat{Z}$. Then the covariance matrix can be decomposed by singular value decomposition: $H = U\Sigma V^T$. The optimal rotation matrix $R$ can be formulated as: $R = UV^T$.

**Protein-Protein Interaction Prediction.** Since the limitation of GPU memory, it is not feasible to input two proteins in the same sequence. Instead, we leverage the representations of proteins to calculate protein-pair attention in decoder. Then the pair-aware protein representations can be obtained by multiplication of protein representations and the attention.

## A.3    ALGORITHMS FOR PROMPT-GUIDED MULTI-TASK PRE-TRAINING AND FINE-TUNING

To more easily appreciate the whole procedure of the prompt-guided multi-task pre-training and fine-tuning framework, we provide pseudo codes as follows.

# B    ADDITIONAL DETAILS OF EXPERIMENTAL SETTING

## B.1    PRE-TRAINING DATASET

To exploit primary structure information, language modeling has been prove effective (Elnaggar et al., 2021; Alley et al., 2019). We follow Rives et al. (2021) to use UniRef50 (Consortium, 2021) dated March 28, 2018. 10% of UniRef50 clusters are randomly selected as a held-out evaluation set.

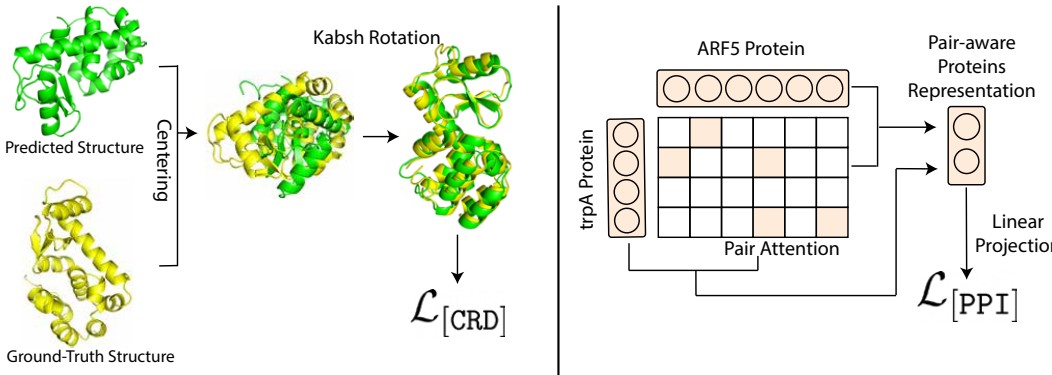

Figure 8: The Overview of Two Additional Pre-training Tasks. **Left**: Alpha-Carbon Coordinate Prediction. **Right**: Protein-Protein Interaction Prediction.

---

**Algorithm 1:** Prompt-Guided Multi-Task Pre-Training

---

**Data:** Input protein $x$, prompt pool $p \in P = \{[\text{MLM}], [\text{CRD}], [\text{PPI}]\}$, task objectives $\mathcal{L}_p$,
    the learning rate $\zeta$.
**Result:** Model parameters $\psi$
  **while** not converge **do**
    **for** $p \in P$ **do**
      Initialize the task-specific input $x_p = x || p$
      Compute the feature $h_p = f_\psi(x_p; \psi)$
        // $f_\psi$ contain $L$ layers Prompt-aware Attention Module
      Compute the loss $\mathcal{L}_p(h_p)$ according to Equation 3, 4 or 5
    **end for**
    Update the model parameters $\psi = \psi - \sum_p (\alpha_p \cdot \zeta \nabla_\psi \mathcal{L}_p)$ according to Equation 6
    Update the prompt parameters $p = p - \alpha_p \cdot \zeta \nabla_p \mathcal{L}_p, \ \forall p \in P$
  **end while**

---

**Algorithm 2:** Prompt-Guided Fine-tuning

---

**Data:** Input protein $x$, downstream task object $\mathcal{L}'_p$,
    learned prompt pool $P = \{[\text{MLM}], [\text{CRD}], [\text{PPI}]\}$, pre-trained model parameters $\psi$,
    the learning rate $\zeta$.
**Result:** Prompt-tuning module parameters $\theta$.
  **while** not converge **do**
    Compute combined prompt $p' = \tau_\theta(p)$ according to Equation 7
    Initialize the input $x_{p'} = x || p'$
    Compute the feature $h_{p'} = f(x_{p'}; \psi)$
    Compute the loss $\mathcal{L}_{p'} = \mathcal{L}_{p'}(h_{p'})$
    Update the prompt-tuning module parameters $\theta = \theta - \zeta \nabla_\theta \mathcal{L}_{p'}$
  **end while**

---

For secondary and tertiary structure information, we extract data from Protein Data Bank (PDB) (Berman et al., 2000). For compatibility with pre-trained protein models, we only use proteins whose amino acid sequence length is less than 1,024. There are many ways to define the coordinates of protein residues. Here we use the coordinates of carbon alpha atoms.

The pre-training dataset for quaternary structures is constructed based on the latest STRING (Szklarczyk et al., 2019) database with only the physical-only mode, which means edges between the protein pairs indicate evidence of their binding or forming a physical complex. The database contains in total 65 million protein sequences from 14,094 species and 2.7 billion protein-protein interaction pairs. Note that there is no edge between proteins that come from different species.

We observe that the PPI network has a problem of uneven distribution, as illustrated in Figure 9, the largest network contains 60,000 proteins and $3.5 \times 10^7$ edges. Such data distributions can lead models to over-focus on proteins from a single species. We pre-process our dataset by choosing the species networks with comparable sizes. Figure 10 illustrates the data distribution after pre-processing.

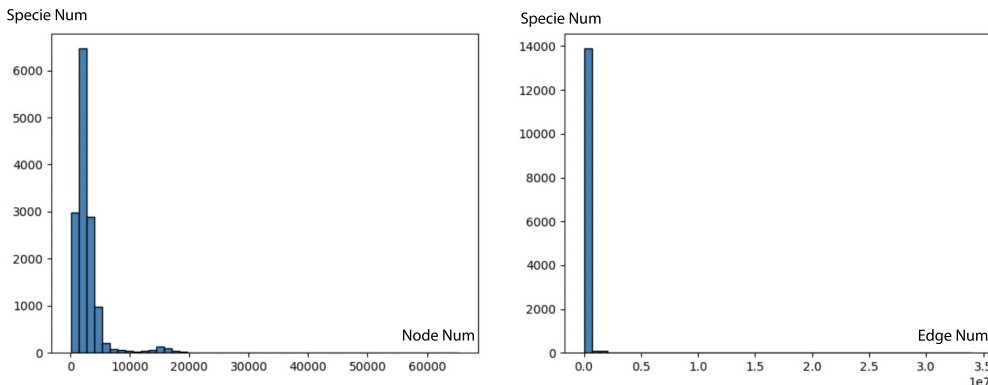

Figure 9: Visualization of the number of nodes and the number of edges in the original database.

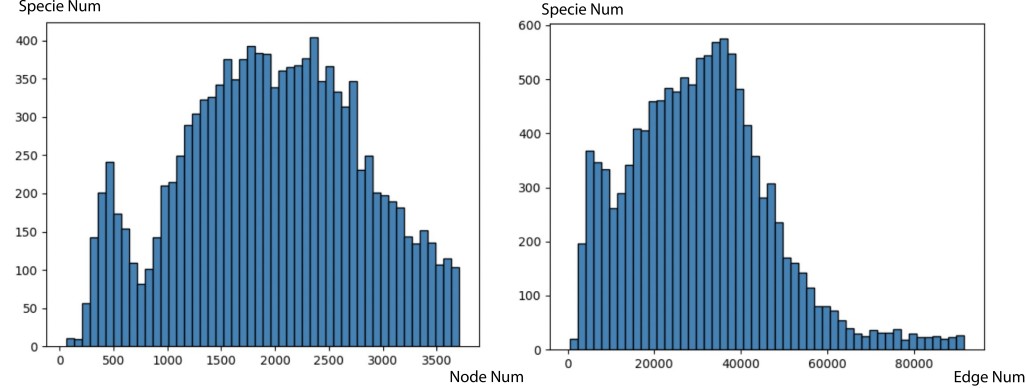

Figure 10: Visualization of the number of nodes and the number of edges in the pre-processed database.

## B.2   DOWNSTREAM TASK DATASETS.

The statistical results of the downstream datasets are shown in Table 4.

**Evaluation Metrics** For multiple binary classification tasks, we employ protein-centric maximum F-score $F_{max}$ and pair-centric area under precision-recall curve $AUPR_{pair}$ to evaluate protein models. For regression tasks, we employ spearman's correlation $\rho$ to evaluate protein models.

Table 4: Statistics of the downstream datasets.

| DATASET | #TRAIN | #VALIDATION | #TEST | TASK |
|---|---|---|---|---|
| ENZYME COMMISSION | 15,551 | 1,729 | 1,919 | CLASSIFICATION |
| GENE ONTOLOGY | 29,902 | 3,323 | 3,416 | CLASSIFICATION |
| STABILITY | 53,679 | 2,447 | 12,839 | REGRESSION |
| FLUORESCENCE | 21,446 | 5,362 | 27,217 | REGRESSION |
| THERMOSTABILITY | 24,817 | - | 3,314 | REGRESSION |
| AAV (1-VS-REST) | 1,170 | - | 81,413 | REGRESSION |
| GB1 (1-VS-REST) | 29 | - | 8,704 | REGRESSION |
| GB1 (2-VS-REST) | 427 | - | 8,306 | REGRESSION |
| SABDAB | 345 | 48 | 99 | REGRESSION |

- $F_{max}$. Given a target protein $i$, we denote its experimentally determined function terms as $T_i$. Given a set of decision threshold $t \in [0,1]$, we denote predicted function terms as $P_i(t)$. The precision and recall of this protein can be formulated as:

$$\text{precision}_i(t) = \frac{\sum_f \mathbb{I}[f \in P_i(t) \cap T_i]}{\sum_f \mathbb{I}[f \in P_i(t)]}, \tag{8}$$

$$\text{recall}_i(t) = \frac{\sum_f \mathbb{I}[f \in P_i(t) \cap T_i]}{\sum_f \mathbb{I}[f \in T_i]}, \tag{9}$$

where $\mathbb{I}[\cdot]$ is an indicator function that is equal to 1 if and only if the condition is true. Combining these two measures, the $F_{max}$ is defined as the maximum value of F-measure:

$$F_{max} = \max_t \{ \frac{2 \cdot \text{precision}(t) \cdot \text{recall}(t)}{\text{precision}(t) + \text{recall}(t)} \}, \tag{10}$$

where $\text{precision}(t) = \frac{1}{M(t)} \sum_i \text{precision}_i(t)$, and $\text{precision}(t) = \frac{1}{N} \sum_i \text{recall}_i(t)$. The $N$ is denoted as the number of proteins and $M(t)$ is denoted as the number of proteins on which at least one prediction result is above threshold $t$.

- $AUPR_{pair}$. The pair-centric area under precision-recall curve is exactly the micro average precision score where precision and recall are for each term $f$:

$$\text{precision}_f(t) = \frac{\sum_i \mathbb{I}[f \in P_i(t) \cap T_i]}{\sum_i \mathbb{I}[f \in P_i(t)]}, \tag{11}$$

$$\text{recall}_f(t) = \frac{\sum_i \mathbb{I}[f \in P_i(t) \cap T_i]}{\sum_i \mathbb{I}[f \in T_i]}. \tag{12}$$

- $\rho$. Spearman's is a nonparametric measure of rank correlation for ground-truth $Y$ and predicted $\hat{Y}$ landscape. We denote $R(\grave{)}$ as ranks. The correlation coefficient is:

$$\rho = \frac{\text{cov}(\text{R(Y)}, \text{R}\hat{Y}))}{\sigma_{R(Y)} \sigma_{R(\hat{Y})}}, \tag{13}$$

where $\text{cov}(\cdot, \cdot)$ is the covariance of the variables, and $\sigma_{R(\cdot)}$ is the standard deviations of the rank variables.

**Enzyme Commission and Gene Ontology.** EC numbers are selected from the third and fourth levels of the EC tree, forming 538 binary classification tasks. GO terms are hierachically organized into three ontologies – biological process (1943 binary classification tasks), molecular function (489 binary classification tasks), and cellular component (320 binary classification tasks). Following DeepFRI (Gligorijević et al., 2021), we use the protein sequences in the test set with 95% sequence identity to the training set.

**Stability Landscape Prediction (Rocklin et al., 2017).** This is a regression task that maps each protein to a label, measuring the most extreme case where the protein maintains its fold above a concentration threshold. This task aims to test the ability to generalize from a broad sampling of

relevant sequences to local neighborhood of a few sequences. The train set includes proteins from experimental design, while the test set contains single mutants.

**Fluorescence Landscape Prediction (Sarkisyan et al., 2016).** This is a regression task that maps a protein to a label corresponding to the log-fluorescence intensity. This task aims to test the ability to distinguish mutants. The train set includes triple mutants of the wild-type green fluorescent protein (GFP), while the test set contains more mutants.

**Thermostability Landscape Prediction (Jarzab et al., 2020).** This is a regression task that maps a protein to a thermostability label. We adopt the mixed split proposed by Dallago et al. (2021) that using MM-seqs2 (Steinegger & Söding, 2017) at a threshold of 20% sequence identity creates one split. The train set includes all sequences in 80% of clusters, while the test contains the remaining 20% of clusters.

**Adeno-associated virus (AAV) Landscape Prediction (Bryant et al., 2021).** This is a regression task that predicts fitness for a long mutated sequence. We adopt the 1-vs-rest split, where wild type and single mutants are assigned to train set, while test set contains the rest. This split are common in protein engineering application.

**GB1 Landscape Prediction (Wu et al., 2016).** This is a regression task that predicts the effects of interactions between mutations. We adopt the 1-vs-rest (and 2-vs-rest) split, where wild type and single mutants (and double mutants) are assigned to train set, while test set contains the rest.

**Antibody-antigen Affinity Prediction (Dunbar et al., 2014).** This is a regression task that takes a pair of proteins as input and predicts the affinity between them. We adopt random split which contains 493 pairs, 431 antibodies and 401 antigens.

Table 5: Model performance on FLIP benchmark.

| DATASET | THERMO | | AAV | | GB1 | | | |
| | MIXED | HUMAN | HUMAN-CELL | 1-VS-R | 2-VS-R | 1-VS-R | 2-VS-R | 3-VS-R | LOW-VS-HIGH |
| --- | --- | --- | --- | --- | --- | --- | --- | --- | --- |
| ESM-1B | 0.68 | 0.70(0.691) | 0.75(0.673) | 0.04 | 0.26 | 0.32 | 0.36 | 0.54 | 0.13 |
| **OURS** | 0.683 | 0.702 | 0.684 | 0.551 | 0.595 | 0.403 | 0.550 | 0.783 | 0.294 |

To illustrate the advantage of prompt-tuning in low-resource scenarios, we only selected a subset of tasks in the FLIP benchmark. In Table 5, we report the performance of our model on other tasks. Note that, although we use the reported results of esm-1b in the above table, we additionally provide the reproduced results of esm-1b on Thermo(Human) and Thermo(Human-cell). These values (surrounded by brackets) are lower than reported.

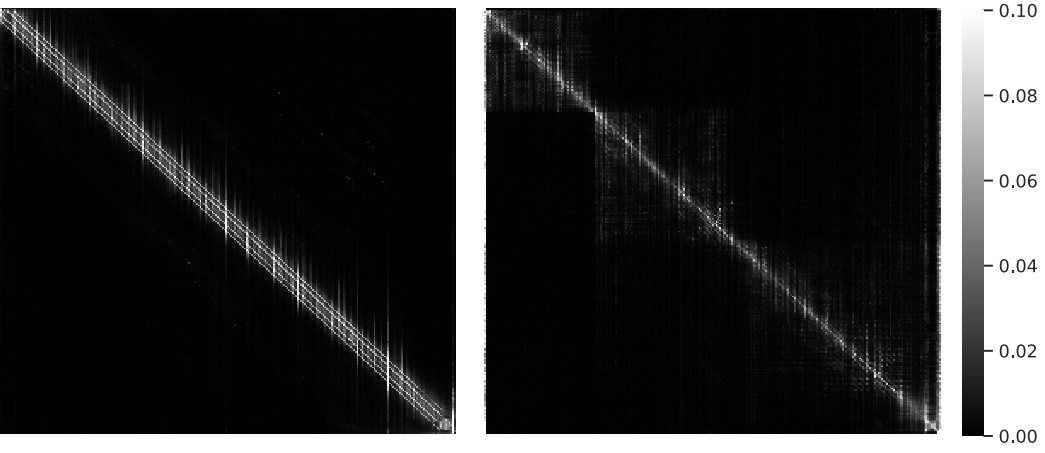

Figure 11: Attention visualization. We select GB1 protein as an example and visualize attentions of the 15-th layer (high skip weight for [MLM]) and the 33-th layer (high skip weight for [CRD] and [PPI]).

## B.3 ANALYSIS OF NEURAL LAYERS

In Figure 11(a), we visualize the attentions in the 15-th layer (a high skip weight for [MLM]) and the 33-th layer (a high skip weight for [CRD] and [PPI]). We observe that one amino acid in the 15-th layer can only attend to the local neighbors in the sequence, whereas the amino acid in the 33-th layer can attend to those amino acids that are more distant along the sequence but potentially closer in the 3D space. This observation demonstrates the primary structural knowledge learned by MLM pays more attention to sequential dependency, whereas the tertiary structural and quaternary structural knowledge learned by CRD and PPI tasks can capture the information from adjacent amino acids in the 3D space.

## B.4 ADDITIONAL EXPERIMENT RESULTS

**Do downstream tasks benefit from the acquired information on-demand by prompt tuning?**

To further analyze the importance of prompt-guided fine-tuning, we conduct an ablation study on the binding affinity prediction task on the SAbDab dataset (Dunbar et al., 2014). From Figure 12, we observe that PromptProtein performs worst without any prompt tokens. In contrast, adding either of the three prompt tokens, especially the token corresponding to the PPI task, can significantly improve performance. By combining different prompts without prompt tuning, we can obtain protein representations enhanced by multiple structural information. By doing that, we find the combination of the [MLM] and [PPI] prompts empowers PromptProtein to achieve the best performance. It is also notable that, by comparing the results of adding [MLM] and [PPI] prompts and adding all prompts, the [CRD] prompt leads to a performance decrease. These results evidence that not all structure information from pre-training is beneficial for downstream tasks, and adaptively combining acquired information via prompt-tuning leads to better performance.

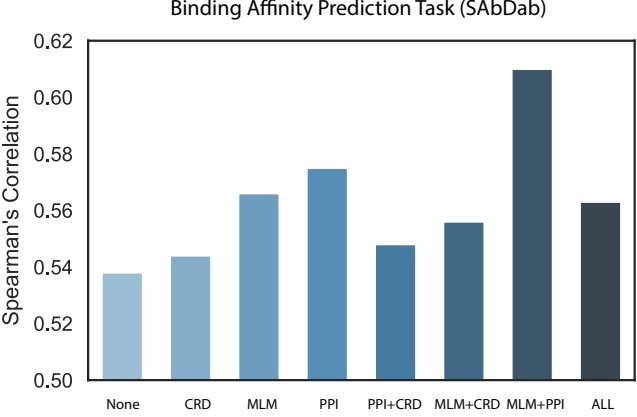

Figure 12: Ablation of PromptProtein with different prompt tokens on SAbDab (spearman's $\rho$ ).

