# OpenReview forum: "Multi-level Protein Structure Pre-training via Prompt Learning"
_ICLR.cc/2023/Conference — ICLR 2023 poster_

### Official Review · Reviewer_iqu6 · 2022-10-23

**Confidence:** 3
**Correctness:** 4
**Technical Novelty And Significance:** 3
**Empirical Novelty And Significance:** 3
**Recommendation:** 6

**Clarity, Quality, Novelty And Reproducibility:**

The paper is well structured. If there are points I missed, I believe it is my inadequacy in biological knowledge.

**Strength And Weaknesses:**

Strengths

1. The paper is well organized and written.
2. Some NLP techniques including pre training and prompt tuning are applied to protein structure study.
3. Empirical study shows good results.

Weaknesses

I would not say this to be a weakness.  The authors lend key techniques from NLP, but protein structure has four distinct levels.
Which levels are being helped by NLP techniques, and which levels are not?

**Summary Of The Paper:**

This paper proposes a prompt-guided multi-task pre-training and fine-tuning framework for protein function prediction.  The authors borrow the techniques from NLP and successfully applied to two tasks and show promising performance.

**Summary Of The Review:**

The work successfully transfers NLP techniques to protein study.  The choice of the techniques including prompt-guided multi-task pre-training and fine-tuning are tailored to apply onto function prediction and protein engineering.  The work is thorough and thought provoking.

---

> ### Author Response · Authors · 2022-11-18
> **Response to Reviewer iqu6**
>
> We would like to express our great appreciation to Reviewer iqu6 for their positive comments on our paper. And our response is as follows,
>
> > *I would not say this to be a weakness. The authors lend key techniques from NLP, but protein structure has four distinct levels. Which levels are being helped by NLP techniques, and which levels are not?*
>
> To learn the primary structural (i.e., sequential) information, we borrow the MLM objective, and design two pre-training objectives to learn the tertiary and quaternary structural information. And, to equip language models with the ability to flexibly mix the acquired information on-demand, we borrow the concept of prompt learning from NLP. Note that, NLP researchers leverage prompts in fine-tuning to improve model performance. In contrast, we extended this concept to the pre-training stage and designed a prompt-guide pre-training and fine-tuning framework, which can not only decouple multiple pre-training tasks but flexibly mix the acquired information on-demand.

---

### Official Review · Reviewer_rpyU · 2022-10-24

**Confidence:** 4
**Correctness:** 4
**Technical Novelty And Significance:** 3
**Empirical Novelty And Significance:** 3
**Recommendation:** 6

**Clarity, Quality, Novelty And Reproducibility:**

Clarity: Overall satisfying, with only a few technical details missing.

Quality: Good.

Novelty: The prompt-based multi-task learning framework is novel for protein structure pre-training.

Reproducibility: Good if detailed design of structure predictor and prompt tuning modules can be further explained.

**Strength And Weaknesses:**

Pros:
1. The overall motivation is well founded and executed. Combining different level of supervised information for protein structure pre-training indeed leads to more informative embeddings, and the prompt-based multi-task learning avoids possible interference between different tasks. The visualization in Figure 4 and 5 provides clear evidence to support the proposed learning framework.
2. Extensive experiments are conducted on two types of down-stream tasks, function annotation and protein engineering. Multiple benchmark datasets are used, and many closely-related baselines are included for comparison, which makes the empirical evaluation hightly convincing.

Cons:
1. It seems that some details are missing (please correct me if I failed to find them in the manuscript). This includes the design of structure predictor (a MLP network?), and the prompt tuning module (linear combination of MLM/CRD/PPI prompts with learnable coefficients?).
2. In Equation (2), it looks like that the linear projection of prompt is layer independent, i.e., the weight is constant in all the layers, which conflicts with Figure 4. Please clarify.
3. Currently, the CRD task corresponds to the second and third levels of protein structures. Since the second level of protein structures refers to secondary structures (alpha-helix and beta-sheet), it may be more natural to introduce secondary structure prediction as learning task for this level. The training data can be the same as the CRD task, but with different supervised information.
4. Authors mentioned that the average error of coordinate prediction task on a single residue is 5A, which may indicate that the model is well trained to minimize this loss. This could be caused from the relatively simple design of structure predictor (if it is a MLP network), which I assume may not work well if the amino-acid sequence is long (e.g. more than 100 amino-acid residues). It should be helpful to use more sophisticated structure predictors (e.g. AlphaFold2’s structure module), but this may be out of the scope of this paper.

**Summary Of The Paper:**

In this paper, authors propose a novel prompt-guided multi-task pre-training and fine-tuning framework for protein structure pre-training. Multi-level supervised information, including masked language modeling (MLM), CA coordinate prediction (CRD), and protein-protein interaction (PPI), are integrated into one unified learning framework without interfering with each other. Extensive empirical evaluation and embedding visualization validate the effectiveness of the proposed method.

**Summary Of The Review:**

The proposed prompt-based learning framework for protein structure pre-training is novel, and indeed resolves the possible interference between different learning tasks. A few minor issues need to be resolved in the author feedback to further improve the overall quality.

---

> ### Author Response · Authors · 2022-11-18
> **Response to Reviewer rpyU**
>
> We thank Reviewer rpyU for reviewing our paper and providing thoughtful feedback on our work. We have revised the manuscript to make this paper easier to follow. Here, we provide details on comments below.
>
> > *It seems that some details are missing (please correct me if I failed to find them in the manuscript). This includes the design of structure predictor (a MLP network?), and the prompt tuning module (linear combination of MLM/CRD/PPI prompts with learnable coefficients?).*
>
> In this paper, we use a simple MLP network as the structure predictor, and a linear layer to combine MLM/CRD/PPI prompt tokens as our prompt tuning module.
>
> > *In Equation (2), it looks like that the linear projection of prompt is layer independent, i.e., the weight is constant in all the layers, which conflicts with Figure 4. Please clarify.*
>
> Sorry for the confusion, the layer gate coefficient $\boldsymbol{g}_p$ is not layer independent. In the $l$-th layer, we should replace, in the paper, the symbol of layer gate coefficient $\boldsymbol{g}_p$ with $\boldsymbol{g}_p^{(l)}$, which depends on the embedding of $l$-th layer prompt $p$.
>
> > *Currently, the CRD task corresponds to the second and third levels of protein structures. Since the second level of protein structures refers to secondary structures (alpha-helix and beta-sheet), it may be more natural to introduce secondary structure prediction as learning task for this level. The training data can be the same as the CRD task, but with different supervised information.*
>
> From the view of our biologist co-authors, in the classical setting before the deep learning era, protein secondary structure prediction is typically regarded as an intermediary step in predicting protein tertiary structures. With downstream tasks in mind, we observe from the biological implications that most application scenarios directly correlate with protein tertiary structures instead of secondary structures. As a result, taking the spirit of end-to-end learning leads us to leave alone this intermediary task of secondary structure prediction. In other words, our design of pretraining tasks follows the principle "Simplify Until Nothing Can Be Removed".
>
> > *Authors mentioned that the average error of coordinate prediction task on a single residue is 5A, which may indicate that the model is well trained to minimize this loss. This could be caused from the relatively simple design of structure predictor (if it is a MLP network), which I assume may not work well if the amino-acid sequence is long (e.g. more than 100 amino-acid residues). It should be helpful to use more sophisticated structure predictors (e.g. AlphaFold2’s structure module), but this may be out of the scope of this paper.*
>
> Thanks for providing direction for model improvement. This paper is mainly to explore the application of prompt technology in pre-trained protein models. We leave how to design efficient decoders to effectively learn multilevel structural information for future work.

---

### Official Review · Reviewer_9Vnq · 2022-10-24

**Confidence:** 3
**Correctness:** 3
**Technical Novelty And Significance:** 2
**Empirical Novelty And Significance:** 2
**Recommendation:** 5

**Clarity, Quality, Novelty And Reproducibility:**

While the paper is mostly well written and easy to follow, there are a few low-level details that seem to be lacking from the manuscript. In brief, to the best of my knowledge:

+ Precise architectural details are lacking.
+ The structure prediction module $\kappa$ is not described in detail.
+ The prompt-tuning module $\tau_{\theta}$ is not described in detail.

Other questions:

1. Why does the performance of PromptProtein in Tables 2 and 3 disagree by a factor larger than the reported standard error?
2. Why do some visualisations use t-SNE and others MDS? It would be best to stick to one approach for consistency and avoid risk of cherry picking.
3. Are all prompts of length one? Have you experimented with longer prompts?
4. During fine-tuning, are only the prompt-tuning module parameters learnt, or are the main model parameter's also fine-tuned? (Algorithm 5 would suggest the former, but this could be more clearly stated in the main manuscript).
6. Did you experiment with saturating gates (e.g. sigmoid) for the skip-connection? Does it even occur that $g_P > 1$, implying that the skip-connection contribution changes sign? If so, what's the interpretation?

# Typos:

equivalence -> equivariance
FILP -> FLIP

**Details Of Ethics Concerns:**

NA.

**Strength And Weaknesses:**

Strengths:
+ Good results on function annotation and protein engineering tasks.

Weaknesses:
+ Results shown only for a subset of sub-tasks in the FLIP benchmark.
+ Ablation experiments are insufficient to disentangle the importance of the proposed architectural and task-related changes.
+ Insufficient details for full reproducibility in the manuscript.

Other weaknesses:
+ Limited methodological novelty (not critical for an application-focused paper).

**Summary Of The Paper:**

This paper introduces PromptProtein, a prompt tuning-based protein language model that combines pre-training tasks at the sequence, structure and protein-protein interaction levels. The model is then fine-tuned on function annotation and protein engineering downstream tasks.

In a nutshell, PromptProtein introduces learnable prompt embeddings for each of the three pre-training tasks. During fine-tuning, a small network learns to combine the three task-specific prompt embeddings into a prompt embedding that is specific to the downstream task at hand. The authors also propose two minor changes to the transformer architecture. Namely:
+ Attention from prompt tokens to sequence tokens is masked.
+ A learnable gate that is a linear function of the prompt token is applied to the skip connections.

Results on both function annotation and protein engineering downstream tasks are favourable relative to the baselines.


**Summary Of The Review:**

Within the extremely active field of protein language models, this paper introduces the following changes over the baselines:

1. While many baselines (e.g. ESM) pre-train only on sequences, the proposed approach pre-trains on three different tasks: masked language modelling on sequences, structure prediction and protein-protein interaction prediction.
2. The proposed approach relies on prompt tuning for (1) fine-tuning and (2) to allow for a degree of task specialisation at pre-training time.

While these changes are minor from a purely methodological viewpoint, I believe they are of interest to the protein language modelling literature. Moreover, these innovations appear to translate into improved downstream performance in function annotation and protein engineering tasks.

However, this work is not without limitations. Most notably, in my opinion, the experiments currently have two main shortcomings:

1. For most of the FLIP tasks (Thermostability, AAV, GB1), the authors show results only for a small subset of task variants. Instead, performance should be shown for *all* of the variants in the FLIP benchmark to prevent any potential cherry-picking.
2. The ablations do not provide sufficiently compelling evidence that the performance improvements are due to the specifics of PromptProtein and not e.g. the use of prompt tuning in general.

To this end, I recommend:
1. Providing experimental results for all subtasks in the FLIP benchmark.
2. Extend the ablation study to all tasks instead of GB1 (2-vs-rest) only.
3. To study the importance of pre-training with different prompt tokens for each task, compare to an ablation baseline that has a single learnable prompt of length three that is shared across the three pre-trained tasks, as opposed to three task-specific prompts of length one. This would help understand to which extent the performance improvement may be simply due to the extra expressivity in the skip-connections or, most importantly, using prompt tuning for fine-tuning to the downstream tasks as opposed to e.g. full model fine-tuning.
4. To study the importance of multi-task pre-training, results should be shown for ablation baselines trained without each of the three tasks, one at a time.

Because of this, I slightly lean towards weak rejection of the manuscript for now, but strongly encourage the authors to address these issues in the rebuttal.

---

> ### Author Response · Authors · 2022-11-18
> **Response to Reviewer 9Vnq (1/2)**
>
> We would like to express our great appreciation to reviewer 9Vnq for their comments on our paper. The typo and unclear content have been revised in the manuscript, and the response to your concerns is as follows:
>
> > *Results shown only for a subset of sub-tasks in the FLIP benchmark.*
>
> > *For most of the FLIP tasks (Thermostability, AAV, GB1), the authors show results only for a small subset of task variants. Instead, performance should be shown for all of the variants in the FLIP benchmark to prevent any potential cherry-picking.*
>
> > *Providing experimental results for all subtasks in the FLIP benchmark.*
>
> To illustrate the advantage of prompt-tuning in low-resource scenarios, we only selected a subset of sub-tasks in the FLIP benchmark. We try our best to fulfill the requirements of Reviewer 9Vnq, while as [1] described, large pre-trained models require amounts of compute (up to 50 days on an NVidia A6000 GPU) to train on some tasks, which is out of the reach of most academic research groups, there are several sub-tasks (AAV: Mut-Des, Des-Mut, 7-vs-rest, low-vs-rest) that we cannot reproduce during the limited rebuttal time. All available results we can obtain are reported below, and the others will be reported later.
>
> | Dataset | GB1 (1-vs-rest) | GB1 (2-vs-rest) | GB1(3-vs-rest) | GB1(low-vs-high) | AAV (1-vs-rest) | AAV (2-vs-rest) | Thermo (Mixed) | Thermo (Human) | Thermo (Human-Cell) |
> | :-----| :----: | :----: | :----: | :----: | :----: | :----: | :----: | :----: | :----: |
> | ESM-1b | 0.32 | 0.36 | 0.77 | 0.13 | 0.04 | 0.26 | 0.68 | 0.70 (0.691) | 0.75 (0.673) |
> | Ours | 0.403 | 0.550 | 0.783 | 0.294 | 0.551 | 0.595 | 0.683 | 0.702 | 0.684 |
>
> Note that, although we use the reported results of esm-1b in the above table, we additionally provide the reproduced results of esm-1b on Thermo(Human) and Thermo(Human-cell). These values (surrounded by brackets) are lower than reported.
>
> > *Ablation experiments are insufficient to disentangle the importance of the proposed architectural and task-related changes.*
>
> > *The ablations do not provide sufficiently compelling evidence that the performance improvements are due to the specifics of PromptProtein and not e.g. the use of prompt tuning in general.*
>
> > *Extend the ablation study to all tasks instead of GB1 (2-vs-rest) only.*
>
> > *To study the importance of pre-training with different prompt tokens for each task, compare to an ablation baseline that has a single learnable prompt of length three that is shared across the three pre-trained tasks, as opposed to three task-specific prompts of length one. This would help understand to which extent the performance improvement may be simply due to the extra expressivity in the skip-connections or, most importantly, using prompt tuning for fine-tuning to the downstream tasks as opposed to e.g. full model fine-tuning.*
>
> > *To study the importance of multi-task pre-training, results should be shown for ablation baselines trained without each of the three tasks, one at a time.*
>
> Thanks for pointing out the deficiencies of the ablation study. Following your advice, we disentangle the architecture and pre-training tasks and fine-tune these models on diverse downstream tasks. We extend to GB1(low-vs-high), AAV(1-vs-rest), and Thermo(Human-Cell) datasets. The results are as follow:
>
> | Dataset | GB1 (low-vs-high) | AAV (1-vs-rest) | Thermo (Human-Cell) |
> | :----- | :-----: | :-----: | :-----: |
> | Conventional MTL | 0.238 | 0.525 | 0.651 |
> | PromptProtein | 0.279 | 0.544 | 0.672 |
> | - Attention Mask | 0.264 | 0.531 | 0.663 |
> | - Layer Gate | 0.270 | 0.520 | 0.659 |
> | - MLM objective | 0.240 | 0.493 | 0.629 |
> | - CRD objective | 0.262 | 0.535 | 0.647 |
> | - PPI objective | 0.253 | 0.532 | 0.654 |
> | Fixed three prompt | 0.242 | 0.523 | 0.646 |
>
> We will replenish the final version with these results.
>
> > *Insufficient details for full reproducibility in the manuscript.*
>
> > *Precise architectural details are lacking. The structure prediction module $\kappa$ is not described in detail. The prompt-tuning module $\tau_\theta$ is not described in detail.*
>
> For the structural prediction module, we use a simple 2-layer MLP head to project amino acid embeddings into the coordinate space. The prompt-tuning module is a linear layer to integrate the three learned prompts. We will clarify this part in the final draft.

---

> > ### Author Response · Authors · 2022-11-18
> > **Response to Reviewer 9Vnq (2/2)**
> >
> > > *Why does the performance of PromptProtein in Tables 2 and 3 disagree by a factor larger than the reported standard error?*
> >
> > In Table 3, we reported the mean $\pm$ standard deviation. Since [1] does not report mean and standard deviation, we suspect they reported their best values and thus we picked our best results for comparison in Table 2.
> >
> > > *Why do some visualisations use t-SNE and others MDS? It would be best to stick to one approach for consistency and avoid risk of cherry picking.*
> >
> > While t-SNE and MDS aim to reduce the data dimensionality for visualization, their goals are different [2]. The t-SNE focuses on maintaining neighborhood data points, while MDS aims to preserve the distances between pairwise data, focusing on pairs of distant points in the original space. In Fig. 5, we aim to visualize the clustering patterns caused by different pre-training methods. Thus, using t-SNE is more appropriate. In another case, we need to compare the whole distribution between the MLM prompt-enhanced representations and the PPI prompt-enhanced representations, which means the MDS dimensionality reduction method is more suitable.
> >
> > > *Are all prompts of length one? Have you experimented with longer prompts?*
> >
> > Yes, the length of all prompts is one, and we did not experiment with longer prompts. The focus of our paper is to use prompt technology to guide the model to achieve multi-task learning without information interference.
> >
> > > *During fine-tuning, are only the prompt-tuning module parameters learnt, or are the main model parameter's also fine-tuned? (Algorithm 5 would suggest the former, but this could be more clearly stated in the main manuscript).*
> >
> > Yes, we only tune the prompt-tuning module and decoder parameters.
> >
> > > *Did you experiment with saturating gates (e.g. sigmoid) for the skip-connection? Does it even occur that $g_p \gt 1$, implying that the skip-connection contribution changes sign? If so, what's the interpretation?*
> >
> > We have experimented with saturating gates (sigmoid) for skip-connection. However, the saturating gates caused the problem of gradient explosion and loss convergence difficulty. Thus, we decided not to utilize saturating gates. Our model did not occur that $\boldsymbol{g}_p > 1$.
> >
> > ---
> > [1] Dallago C, Mou J, Johnston K E, et al. FLIP: Benchmark tasks in fitness landscape inference for proteins[C]//Thirty-fifth Conference on Neural Information Processing Systems Datasets and Benchmarks Track (Round 2). 2021.MLA
> >
> > [2] Manifold learning.https://scikit-learn.org/stable/modules/manifold.html

---

### Official Review · Reviewer_M4p8 · 2022-10-30

**Confidence:** 4
**Correctness:** 3
**Technical Novelty And Significance:** 3
**Empirical Novelty And Significance:** 3
**Recommendation:** 6

**Clarity, Quality, Novelty And Reproducibility:**

**Clarity**
- Could you please clarify how you handle the instances where not all tasks are available for the same proteins in a given mini-batch (eg., instances where primary structure is available in Uniref50, but structure is not available in the PDB)?
- I would clarify in section 3.1 that $g_p$ (the linear projection of the prompt p) is a scalar. In my first lecture, I had (wrongly) assumed this was a vector of the same size as $h_p$ but that was not really making sense anymore when reading the analysis in section 4.
- “We observe that one amino acid in the 15-th layer can only attend to the local neighbors in the sequence, whereas the amino acid in the 33-th layer can attend to those amino acids that are more distant along the sequence but potentially closer in the 3D space” (in section B.3) —> I found this particularly insightful and would recommend to move this section from supplementary to the main text if space allows.
- What is the nature of the operator $\tau_\theta$ used for fine tuning? Linear projection? Any non-linearity applied?
- What is ESM-unstrained (Table 2)?
- Fig5 analysis — it is not clear what is being done here. Is this analysis conducted for a particular protein sequence (if yes, which one)? Or some aggregation over Uniref50 sequences? The embeddings from which layer(s) are being used here?
- I find the “skip connection” terminology a bit confusing as it seems that the term $g_p$ is used as a multiplicative factor for the attention-based transform but not the skip connection term. Also on Figure 3, the arrow labeled “skip connection” is in fact not the skip connection as it supports the computation of the  $g_p$ term which should not be present in the skip connection as per equation 2.
- In the conclusion: “In the future, we are interested in examining the effectiveness of the proposed prompt-guided multi-task pre-training and fine-tuning framework in domains where hierarchical task information is required in the pre-training stage.” —> Could you please provide examples for such domains?

**Quality**
- Very sound approach overall. Authors provided some very compelling empirical results, yet there are a few concerns with some of the ablation results as detailed above. Also the SOTA claim in the is not fully substantiated as discussed above as well.

**Novelty**
- The prompt masking and skip-connection weights are novel to my knowledge and appear to be both critical to the strong reported performance. Could the authors please confirm these two ideas are indeed introduced for the first time in this paper and not borrowed from the NLP literature?

**Reproducibility**
- Authors confirm that the code will be open sourced upon acceptance (section 4.1)


**Strength And Weaknesses:**

**Strengths**
- The idea of combining different pre-training tasks seems very sensible for proteins given the diversity of modalities that characterize them (eg., via their primary, secondary, tertiary and quaternary structure). As noted by the authors, there is however a high risk of task interference when one wants to obtain pre-trained representations that combine these different modalities / tasks. The approach suggested by authors appears to be doing a fine job at efficiently combining these different modalities given the performance reported in sections 4.2. and 4.3.
- The introduced prompt masking and skip connections both appear to be critical to strong empirical performance — the latter seems to be particularly important to mitigate task interference as evidenced by the ablations in section 4.4.
- The paper is very clear overall (in particular the methodology section) with nice visuals facilitating understanding and additional analyses in section 4.5 to help investigate the source of the performance lift.

**Weaknesses**
- The ablations for the different pre-training tasks in section 4.5 / Figure 6 are a bit puzzling. It does seem that the CRD task has destructive value on that particular binding affinity prediction task since: a) the performance of CRD + MLM or CRD + PPI leads to both lower performance Vs MLM or PPI alone respectively b) the performance of CRD + MLM + PPI is also lower vs just using MLM + PPI. This seems particularly important from a practical standpoint, and additional experiments are needed to confirm whether: 1) that problem applies to other downstream tasks or is just specific to binding affinity prediction — and if so, why?  2) there is something fundamentally wrong with the CRD pre-training as currently implemented? 3) there is a way to anticipate ex ante (or post fine tuning) which tokens should be used to ensure optimal task performance ?
- The ablation in section in Table 3 is a bit puzzling as well: it appears that the performance of PromptProtein without layer skip is lower than the performance from the conventional MTL. Could you please explain why that might be the case? (I would have assumed intermediate performance between conventional MTL and full PromptProtein as I presume the attention masks are still used in that ablation?)
- Several points (in section 4 primarily) were not fully clear (see clarity paragraph below).
- The following claim in conclusion does not seem fully substantiated: “PromptProtein beats state-of-the-art baselines by significant margins”. Authors do report the relevant baselines listed in the FLIP paper [1]. But since that paper was released, several methods have shown markedly superior performance for protein modeling & achieving high spearman with deep mutational scanning assays — see for example, [2] and [3]. I would suggest adding these two baselines to the analysis or tone done the SOTA claims.


------------------------------------------------------------------------------------------------------------------------
[1] Dallago, C., Mou, J., Johnston, K.E., Wittmann, B.J., Bhattacharya, N., Goldman, S., Madani, A., & Yang, K.K. (2022). FLIP: Benchmark tasks in fitness landscape inference for proteins. bioRxiv.

[2] Hsu, C., Verkuil, R., Liu, J., Lin, Z., Hie, B.L., Sercu, T., Lerer, A., & Rives, A. (2022). Learning inverse folding from millions of predicted structures. bioRxiv.

[3] Notin, P., Dias, M., Frazer, J., Marchena-Hurtado, J., Gomez, A.N., Marks, D.S., & Gal, Y. (2022). Tranception: Protein Fitness Prediction with Autoregressive Transformers and Inference-time Retrieval. ICML.


**Summary Of The Paper:**

This paper proposes a prompt-based multi-task framework for pre-training and fine-tuning protein sequence representations. From a methodological standpoint, the paper adapts the prompt fine-tuning idea (ie., a differentiable continuous prompt token is pre-trained instead of using a discrete prompt) from the NLP literature to protein modeling with large transformer networks. It then makes two contributions over the standard transformer architecture 1) A specific masking scheme for the prompt token is used to keep only the effect of the prompt on the input sequence and eliminate the opposite effect (ie., the prompt should be task-dependent and not sample-dependent) 2) Learned task-specific layer-specific skip connection linear projections to let the network learn different weights for each task at each layer. The multi-task pre-training involves three kinds of pre-training: a) masked-language modeling with sequence-based information only (MLM) b) Prediction of the alpha-carbons positions (CRD) c) prediction of protein-protein interactions (PPI). Experiments on function annotation and protein engineering (FLIP benchmark) demonstrate the benefits of the different ideas introduced in this work.

**Summary Of The Review:**

This paper is aiming to address a very important area in protein modeling: learning rich sequence embeddings by leveraging multiple pre-training tasks jointly. The approach is sound and the methodological section is overall very clearly presented. There are a few concerns with respect to some of the results and claims as detailed above. Given the several strengths of the work, I would be willing to increase my score if authors address these points during rebuttal.

------------------------------------------------------------------------------------------------------------------------------------------------------------
[UPDATE POST REBUTTAL]

My most important concerns have been alleviated during rebuttal and I have increased my score accordingly.

---

> ### Author Response · Authors · 2022-11-18
> **Response to Reviewer M4p8 (1/2)**
>
> We thank Reviewer M4p8 for reviewing our paper and providing constructive feedback. We believe that appropriately addressing these concerns will make our paper stronger. The following is our response to your concerns.
>
> > *It does seem that the CRD task has destructive value on that particular binding affinity prediction task. This seems particularly important from a practical standpoint, and additional experiments are needed to confirm whether: 1) that problem applies to other downstream tasks or is just specific to binding affinity prediction — and if so, why? 2) there is something fundamentally wrong with the CRD pre-training as currently implemented? 3) there is a way to anticipate ex ante (or post fine tuning) which tokens should be used to ensure optimal task performance?*
>
> Since the CRD prompt ought to drive our model to acquire native conformational information, we conduct additional experiments on contact map prediction and the results are as follow:
>
> | Prompt Combination | MLM | CRD | PPI | MLM + CRD | MLM + PPI | CRD + PPI | ALL | Prompt-Tuning |
> | :----- | :-----: | :-----: | :-----: | :-----: | :-----: | :-----: | :-----: | :-----: |
> | Contact Map (P@L/5) | 0.431 | 0.453 | 0.408 | 0.434 | 0.412 | 0.415 | 0.444 | 0.467 |
> | Binding Affinity ($\rho$) | 0.566 | 0.544 | 0.575  | 0.556 | 0.610 | 0.548 | 0.563 | 0.623 |
>
> For this table, we observe that the CRD prompt has constructive value on contact map prediction, which rejects the hypothesis of something wrong with the CRD pre-training task (problem 2). These two opposite observations are in line with the findings of [1] that structural information is less effective than sequence information on fitness prediction tasks. Thus, we propose to equip the conventional multitask learning approach with the prompt tuning technique, which can flexibly mix the acquired information on-demand.
>
> > *The ablation in section in Table 3 is a bit puzzling as well: it appears that the performance of PromptProtein without layer skip is lower than the performance from the conventional MTL. Could you please explain why that might be the case? (I would have assumed intermediate performance between conventional MTL and full PromptProtein as I presume the attention masks are still used in that ablation?)*
>
> To investigate the above-mentioned problem, we extended this ablation study to gb1 (low-vs-high), aav (one-vs-many), and thermo (human-cell) datasets. By comparing the results of different models on different datasets, we found that the performance of PromptProtein variant without layer gate was inferior to the conventional MTL in low-resource scenarios. We suspect this is because simply adding an attention mask does not adequately decouple different levels of structural knowledge. Especially in low-resource scenarios, it is difficult to flexibly mix information on-demand through prompt-tuning, and adding additional trainable parameters increases the risk of overfitting.
>
> > *Could you please clarify how you handle the instances where not all tasks are available for the same proteins in a given mini-batch (eg., instances where primary structure is available in Uniref50, but structure is not available in the PDB)?*
>
> Since the key idea of our method is to enable a pre-trained model to produce task-specific representations of the same protein in various tasks, we believe it is not compulsory to align three databases for the three pre-training tasks. In practice, we sampled different proteins from these databases to make a mini-batch, so that the model is supposed to simultaneously acquire different task information.
>
> > *I would clarify in section 3.1 that $\boldsymbol{g}_p$ (the linear projection of the prompt $p$) is a scalar. In my first lecture, I had (wrongly) assumed this was a vector of the same size as $h_p$ but that was not really making sense anymore when reading the analysis in section 4.*
>
> Thanks for pointing out this misleading part, we have revised our manuscript to make the symbol $\boldsymbol{g}_p$ clearer.
>
> > *“We observe that one amino acid in the 15-th layer can only attend to the local neighbors in the sequence, whereas the amino acid in the 33-th layer can attend to those amino acids that are more distant along the sequence but potentially closer in the 3D space” (in section B.3) —> I found this particularly insightful and would recommend to move this section from supplementary to the main text if space allows.*
>
> We agree to and will move this section from supplementary to the main text in the final version if space allows.
>
> > *What is the nature of the operator $\tau_\theta$ used for fine tuning? Linear projection? Any non-linearity applied?*
>
> The prompt-tuning module is a linear layer to integrate the three learned prompts.
>
> > *What is ESM-untrained (Table 2) ?*
>
> Following [1], ESM-untrained uses the ESM architecture without pre-training.

---

> > ### Author Response · Authors · 2022-11-18
> > **Response to Reviewer M4p8 (2/2)**
> >
> > > *Fig5 analysis — it is not clear what is being done here. Is this analysis conducted for a particular protein sequence (if yes, which one)? Or some aggregation over Uniref50 sequences? The embeddings from which layer(s) are being used here?*
> >
> > Figure 5 aims to illustrate that different pre-training manners will produce different representation patterns. This analysis is conducted for a particular protein sequence (PDB id: 5JDB for Seq and 1XXO for CRD), and the embeddings come from the last layer. We can observe that, with the MLM prompt guiding, the amino acid embeddings are clustered by the type of amino acids, while, with the CRD prompt guiding, these embeddings are clustered by the positions. The embeddings learned by the conventional MTL attempt to merge multiple structural features at the same time, which leads to an unclear pattern.
> >
> > > *I find the “skip connection” terminology a bit confusing as it seems that the term $g_p$ is used as a multiplicative factor for the attention-based transform but not the skip connection term. Also on Figure 3, the arrow labeled “skip connection” is in fact not the skip connection as it supports the computation of the $g_p$ term which should not be present in the skip connection as per equation 2.*
> >
> > The skip connection term comes from our expectation that the prompt can guide the model to use information from some certain layers. In the implementation process, it is more similar to using a prompt to assign a weight to each layer to achieve the coexistence of multiple knowledge. Thus, it might be more appropriate to call it a skip modulator? We sincerely welcome any more suitable terms to make the article clearer.
> >
> > > *The following claim in conclusion does not seem fully substantiated: “PromptProtein beats state-of-the-art baselines by significant margins”. Authors do report the relevant baselines listed in the FLIP paper [1]. But since that paper was released, several methods have shown markedly superior performance for protein modeling & achieving high spearman with deep mutational scanning assays — see for example, [2] and [3]. I would suggest adding these two baselines to the analysis or tone done the SOTA claims.*
> >
> > During the limited rebuttal period, we have been suggested by reviewers to conduct some time-consuming pre-training experiments. With full respect to your concerns, we will tone done the SOTA claims in our manuscript.
> >
> > > *The prompt masking and skip-connection weights are novel to my knowledge and appear to be both critical to the strong reported performance. Could the authors please confirm these two ideas are indeed introduced for the first time in this paper and not borrowed from the NLP literature?*
> >
> > To the best of our knowledge, these two components (prompt masking and skip-connection weights) are proposed for the first time and not borrowed from the NLP literature.
> >
> > > *In the conclusion: “In the future, we are interested in examining the effectiveness of the proposed prompt-guided multi-task pre-training and fine-tuning framework in domains where hierarchical task information is required in the pre-training stage.” → Could you please provide examples for such domains?*
> >
> > As [2] reported, there is a negative transfer across different task families (e.g. DLG -> SUM). We suspect that the pre-training tasks corresponding to different task families can mine different aspects of semantic information. And we are interested in applying this framework to learn multiple semantic information in the NLP field.
> >
> > ---
> >
> > [1] Hu M, Yuan F, Yang K K, et al. Exploring evolution-aware &-free protein language models as protein function predictors[J]. NeurIPS.
> >
> > [2] Aribandi V, Tay Y, Schuster T, et al. ExT5: Towards Extreme Multi-Task Scaling for Transfer Learning[C]//International Conference on Learning Representations. 2021.

---

> > > ### Comment · Reviewer_M4p8 · 2022-12-12
> > > **Re: responses from authors**
> > >
> > > Dear authors,
> > >
> > > Thank you for the thorough responses during the rebuttal which alleviated my most important concerns -- I have increased my score accordingly.
> > >
> > > A couple final thoughts / suggestions based on your responses:
> > >
> > > 1. Your first two responses were quite insightful and would suggest to include them in appendix
> > >
> > > 2. Re: Fig5 analysis -- I would suggest to clarify that point in the corresponding caption
> > >
> > > 3. Re: skip connection terminology. The skip connection is precisely the bit that is not represented in Fig 3, ie., the part of the architecture that adds $h_p^{(l)}$ to the output of the attention layer in Eq 2. Perhaps you could call $g_p^{(l)}$ an "attention sensitivity" or "attention modulator" (closer to what you suggested)?

---

### Decision · Program_Chairs · 2023-01-20

**Decision:**

Accept: poster

**Justification For Why Not Higher Score:**

The paper translate the prompt related ideas from NLP to the study of proteins. Although the paper proposes some novel modifications to the transformer architecture and presents strong empirical results, the contributions are not ground breaking enough for a spotlight or oral presentation.

**Justification For Why Not Lower Score:**

The paper makes some very useful contributions to the study of proteins and their framework is likely to be adopted by or inspire researchers in the field. Although some of the reviewers did not revise their reviews after rebuttal, I checked thoroughly that their concerns were satisfactorily addressed. The authors did a great job at address them. In particular the new ablation studies and additional experiments are very convincing.

**Metareview: Summary, Strengths And Weaknesses:**

The paper proposes a prompt-guided multi-task pre-training framework and counterpart fine-tuning module to efficiently leverage the various structure levels of a protein.  Experiments on function annotation and protein engineering (FLIP benchmark) demonstrate the value of the approach.

The reviewers and AC all agree that the paper makes some novel and sensible contributions, including the attention mask and skip connection components of the approach, which show very promising results.

The authors did a great job at addressing the reviewers comments. In particular the additional ablation studies and extended experimental results for all subtasks in the FLIP benchmark are great additions to demonstrate the value of the approach.

We strongly encourage the authors to tackle the future work direction on hierarchical task information.

**Note From Pc:**

if the above contains the word "oral" or "spotlight" please see: "oral" presentation means -> notable-top-5% and "spotlight" means -> notable-top-25%. As stated in our emails, we are disassociating presentation type from AC recommendations